# Clinical Characteristics of Mitochondrial DNA Copy Number in Osteonecrosis of the Femoral Head

**DOI:** 10.3390/medicina56050239

**Published:** 2020-05-16

**Authors:** Si-Wook Lee, Kyung-Jae Lee, Beom-Soo Kim, Hyuk-Jun Kwon, Jae-Ho Lee

**Affiliations:** 1Department of Orthopaedic Surgery, Keimyung University School of Medicine, Dongsan Hospital, Daegu 42601, Korea; shuk@dsmc.or.kr (S.-W.L.); oslee@dsmc.or.kr (K.-J.L.); BSKim@dsmc.or.kr (B.-S.K.); purme2000@naver.com (H.-J.K.); 2Department of Anatomy, School of Medicine, Keimyung University, Daegu 42601, Korea

**Keywords:** mtDNA, mtDNA-CN, ONFH, osteonecrosis, mitochondrial DNA, ESR, mtDNA copy number

## Abstract

*Background and objectives*: Alterations in mitochondrial DNA (mtDNA) have been observed and studied in various diseases. However, the clinical value of the mtDNA copy number (mtDNA-CN) alterations in osteonecrosis of the femoral head (ONFH) is poorly understood. In the present study, we investigated whether alterations in mtDNA-CNs are associated with clinicopathological parameters in ONFH. *Materials and methods*: MtDNA-CNs in the synovial tissue of 34 patients with ONFH and 123 control tissues (femoral neck fracture) were measured using quantitative real-time PCR. The present study then analyzed the correlation between the mtDNA-CN and the clinicopathological characteristics of ONFH and fracture patients. *Results*: The average mtDNA-CN (mean ± standard deviation) was 23.82 ± 22.37 and 25.04 ± 24.27 in ONFH and control tissues, respectively, and was not significantly different between the groups (*p* = 0.792). The mtDNA-CN was positively associated with age (27.7% vs. 45.9%, *p* = 0.018) and negatively associated with the erythrocyte sedimentation rate (ESR) (11.8% vs. 39.7%, *p* = 0.024) in all of the samples. The study also found further associations with age (22.2% vs. 68.8%, *p* = 0.014), gender (30.0% vs. 64.3%, *p* = 0.048), and ESR (0% vs. 57.7%, *p* = 0.043) in ONFH. *Conclusions*: in this study, we demonstrated that mtDNA-CN might be a significant marker for predicting clinical characteristics in ONFH.

## 1. Introduction

Human genome sequencing has revealed that human mitochondrial DNA (mtDNA) is composed of 16,569 base pairs, each base pair forming the double helix DNA structure, with multiple copies of mtDNA present in each mitochondrion [1]. They play an essential role in the regulation of cellular metabolism, signaling, and apoptosis, among other functions, but they also have a high mutation rate. Mitochondrial mutations occur 10- to 100-fold more than that of nuclear DNA because of its higher concentrations of reactive oxygen species (ROS), insufficient repair system, and lack of coating proteins, such as the histones in the nucleus [1,2,3]. Genetic studies of mitochondrial diseases demonstrated that a change in mtDNA copy numbers (mtDNA-CN) or mtDNA gene expressions may induce a deficiency in oxidative phosphorylation and the enhanced generation of adenosine triphosphate (ATP) by glycolysis, suggesting the clinical value of mtDNA-CN changes in various diseases [4,5,6].

One such disease caused by the genetic change of mtDNA is osteonecrosis of the femoral head (ONFH). It is a complex multifactorial disease leading to the rapid destruction and dysfunction of the hip joints, and is associated with genetic predisposition and exposure to definite environmental factors [7]. Many etiological factors, such as corticosteroids, alcohol consumption, radiation, and Gaucher disease, are associated with its development [8,9]. About 65–70% of patients with advanced ONFH need total hip replacement. Metabolic disorders have been thought of as a significant factor in ONFH pathogenesis, because steroid use and alcohol consumption are closely associated with ONFH incidence [10]. However, there have been few genetic studies on mtDNA in OFNH and bone disease, while only a few studies have been conducted to investigate its molecular mechanism. To the best of our knowledge, there has been no mtDNA study of synovial tissue performed on ONFH to date, limiting our efforts to clarify the pathogenesis of ONFH. In the present study, we examined the mtDNA-CN in ONFH and control tissues with femoral head fracture patients before analyzing their clinicopathological characteristics.

## 2. Materials and Methods

### 2.1. Patients and DNA Extraction

A total of 157 patients who underwent femoral head surgery for the treatment of ONFH or femoral head fracture at the Keimyung University Dongsan Hospital from September 2009 to October 2011 were contacted. The researchers explained to the patients the study purpose and obtained informed consent from each participant. The Institutional Review Board of the Keimyung University Dongsan Medical Center approved the protocols necessary for the research (IRB No. is DSMC-2016-01-041-001, 05. Feb. 2016). All the ONFH patients were diagnosed by an orthopedic surgeon and classified by the Ficat system as grade III, meaning they were undergoing total hip replacement. The Keimyung Human Bio-resource Bank in Korea provided the synovial tissues, which were immediately frozen in liquid nitrogen and stored at –80 °C until DNA isolation. Synovial tissues was also obtained from the ONFH subjects undergoing total hip replacement within 24 h of a traumatic femoral neck fracture. Patients with severe chronic diseases, such as cardiovascular diseases, congenital diseases, human immunodeficiency virus infection, diabetes mellitus, renal dysfunction, and cancer, were excluded.

Genomic DNA was extracted from the tissue samples using the QIAamp DNA mini kit (Qiagen, Inc., Valencia, CA, USA). The DNA quantity and quality were measured using NanoDrop 1000 (Thermo Scientific, Wilmington, DE, USA).

### 2.2. Mitochondria Copy Number

The MtDNA-CN was analyzed using a real-time quantitative polymerase chain reaction (qPCR) assay. For the quantitative determination of the mtDNA content relative to the nuclear DNA (nDNA), primers for the specific amplification of mtDNA, COX1 and the nDNA-encoded β-actin gene were selected based on a previous study [11]. Real-time qPCR was then carried out on a LightCycler 480 II system (Roche Diagnostics, Germany), with a total volume of 20 µL reaction mixture containing 10 µL SYBR Green Master MIX (Takara, Japan), 8 pmol of each primer, and DNA (50 ng). The PCR conditions were 95 °C for 1 min, followed by 40 cycles of 95 °C for 15 s and 60 °C for 30 s.

Then, the threshold cycle number (Ct) values of the β-actin gene and the mitochondrial COXI gene were determined. The mtDNA-CN in each tested specimen was then normalized against that of the β-actin gene to calculate the relative mtDNA-CN. The present study repeated each measurement in triplicate and included five serially diluted control samples in each experiment.

### 2.3. Statistical Analysis

All the statistical analyses used the SPSS statistical package, version 25.0, for Windows in the present study. The present study presented the mitochondrial copy numbers as the mean ± standard deviation (SD) and considered *p* values < 0.05 to indicate statistically significant results. The present study also used a chi-square test, the Mann–Whitney U test, and a simple correlation analysis to analyze the associations between the variables.

## 3. Results

The mean age of the 157 patients with ONFH or a femoral head fracture was 56.2 years (range, 26–93 years), which was not significantly different in the two groups. There were 70 (44.6%) male patients and 87 (55.4%) female patients. According to the clinical and pathological diagnosis, there were 34 (21.7%) ONFH patients (20 males and 14 females) and 123 (78.3%) femoral head fracture patients (50 males and 73 females).

The researchers analyzed the mtDNA-CN of the femoral tissue using a real-time qPCR. The average mtDNA-CN was 24.77 ± 23.80. The patients were then categorized into two subgroups according to the average value of the mtDNA-CN. The average mtDNA-CN was 23.82 ± 22.37 and 25.04 ± 24.27 in ONFH and femoral head fractures (as the control tissue), respectively. It had no statistical significance (*p* = 0.792), as shown in Figure 1.

The association between the mtDNA-CN changes and the clinicopathological parameters of femoral head injury are summarized in Table 1. Among all the patients, 57 (36.3%) exhibited a higher mtDNA-CN, while 100 (63.7%) exhibited a lower mtDNA-CN. A higher mtDNA-CN was associated with an older age (27.7% vs. 45.9%, *p* = 0.018) and negative erythrocyte sedimentation rate (ESR) (11.8% vs. 39.7%, *p* = 0.024).

The clinicopathological characteristics of mtDNA-CN in ONFH are presented in Table 2. A higher mtDNA-CN was associated with old age (22.2% vs. 68.8%, *p =* 0.014), gender (30.0% vs. 64.3%, *p =* 0.048), and ESR (0% vs. 57.7%, *p =* 0.043). Other clinicopathological characteristics were not associated with mtDNA-CN.

A quantitative analysis showed that mtDNA-CN was positively correlated with age (r = 0.182, *p =* 0.022). When stratified into ONFH and control, this correlation was more significantly found in only ONFH (r = 0.375, *p =* 0.029), as shown in Figure 2. Other variables did not have any association.

## 4. Discussion

To our knowledge, this was the first study to analyze and compare the differences in mtDNA content between tissues from ONFH patients and femur neck fracture patients. We examined the mtDNA-CN in the synovial tissues of ONFH and femur neck fracture patients, as most studies about mtDNA-CN were performed using peripheral blood from patients who have cancer, diabetes, and psychological diseases [3,4,12]. The mtDNA-CN can easily and rapidly change according to the disease pathogenesis [13,14], as mtDNA is prone to oxidative injury. This inclination to oxidative damage is due to the following: (1) mtDNA is not protected by histones, and (2) mitochondria generate reactive oxygen species (ROS) during ATP synthesis. The mtDNA-CN also may be influenced by various factors such as age, gender, obesity, and socio-economic status. Considering these factors, the maintenance of sufficient mtDNA copy contents is necessary for cellular homeostasis, though its level is generally different in each cell [1,2].

As a result, we found that the mtDNA-CN of the ONFH group was similar to that of the control subjects. Interestingly, mtDNA-CN was positively correlated with age only in the ONFH group, and an association between age and mtDNA content showed contradictory results. Previous studies have demonstrated a negative correlation between mtDNA-CN and age [15,16]. However, another study showed that a higher mtDNA-CN was found in younger subjects who were either healthy or suffering from cancer [17]. Notably, a higher mtDNA content in elderly subjects was associated with better clinical characteristics [18,19]. A recent study demonstrated the curvilinear association of mtDNA content with age [20]. Therefore, this association should be confirmed in common and orthopedic diseases in more significant cases.

Interestingly, a lower mtDNA-CN was associated with fluctuations in the erythrocyte sedimentation rate (ESR), which rises in cases of inflammation, infections, anemia, autoimmune disorders, pregnancy, some kidney diseases, and some cancers [21]. Meanwhile, the ESR falls in cases of polycythemia, leukemia, low plasma, sickle cell anemia, hyperviscosity, and congestive heart failure. In ONFH, the ESR is usually elevated by inflammatory processes that are also non-specific [7,9]. It suggested that mitochondrial changes brought on by inflammation may play an important role in ONHF pathogenesis. The researchers suggest that its molecular mechanism should be studied further.

## 5. Conclusions

We identified the clinicopathological characteristics of mtDNA in ONFH tissue. Though the level of mtDNA-CN was not significantly different in ONFH and the control group, its change may contribute to the pathogenesis of ONFH in an age-dependent manner. Further biological studies should clarify the potential mechanism of mitochondrial genes in the development of ONFH.

## Figures and Tables

**Figure 1 medicina-56-00239-f001:**
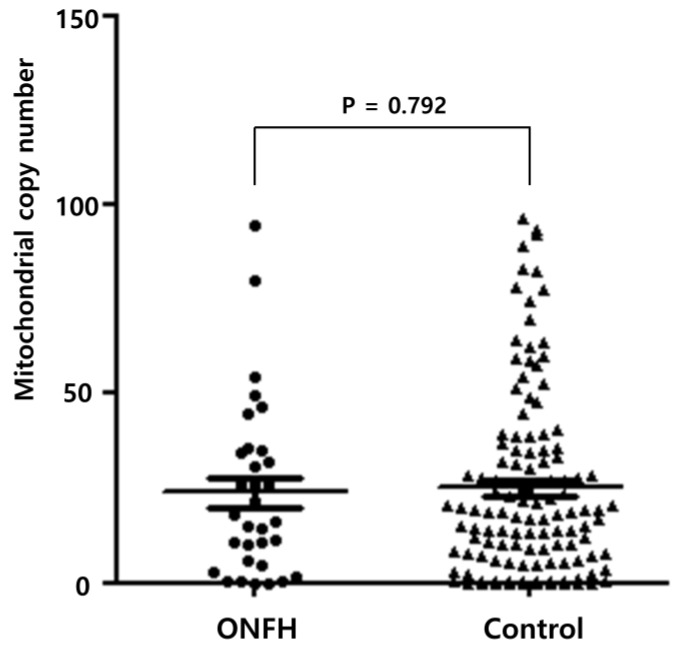
Mitochondrial copy number in osteonecrosis of the femoral head and the control.

**Figure 2 medicina-56-00239-f002:**
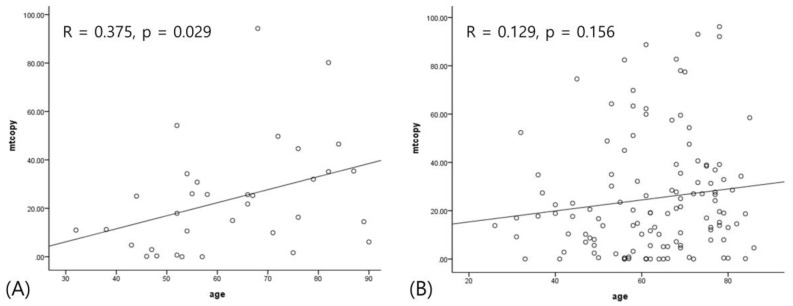
The correlation between age and mitochondrial copy number. (**A**) Positive correlation in osteonecrosis of the femoral head; (**B**) no correlation in fracture.

**Table 1 medicina-56-00239-t001:** Mitochondrial DNA copy number in synovial tissue of femoral head injury.

	mtDNA-CN (%, n)		
	Low	High	*p*
All patients	63.7 (100)	36.3 (57)	
Disease			0.285
ONFH	55.9 (19)	44.1 (15)	
fracture	65.9 (81)	34.1 (42)	
Age			0.018
<65	72.3 (60)	27.7 (23)	
≥65	54.1 (40)	45.9 (34)	
Gender			0.890
Male	64.3 (45)	35.7 (25)	
Female	63.2 (55)	36.8 (32)	
Side			0.533
Right	63.9 (46)	36.1 (26)	
Left	61.8 (42)	38.2 (26)	
Both	80.0 (8)	20.0 (2)	
ESR			0.024
(+)	88.2 (15)	11.8 (2)	
(−)	79 (60.3)	39.7(52)	
CRP			0.471
(+)	55.6 (15)	44.4 (12)	
(−)	63.0 (75)	37.0 (44)	
Alcohol			0.482
(+)	56.5 (13)	43.5 (10)	
(−)	64.2 (79)	35.8 (44)	

**Table 2 medicina-56-00239-t002:** Mitochondrial DNA copy number in the synovial tissue of osteonecrosis of the femoral head (ONFH).

	mtDNA-CN (%, n)
Low	High	*p*
All patients	55.9 (19)	44.1 (15)	
Age			0.014
<65	70.0 (14)	22.2 (4)	
≥65	31.3 (5)	68.8 (11)	
Gender			0.048
Male	70.0 (14)	30.0 (6)	
Female	35.7 (5)	64.3 (9)	
Side			0.430
Right	46.2 (6)	53.8 (7)	
Left	57.1 (8)	42.9 (6)	
Both	80.0 (4)	20.0 (1)	
ESR			0.043
(+)	100 (5)	0 (0)	
(−)	42.3 (11)	57.7(15)	
CRP			1.00
(+)	66.7 (2)	33.3 (1)	
(−)	50.0 (14)	50.0 (14)	
Alcohol			1.00
(+)	55.6 (10)	44.4 (8)	
(−)	50.0 (6)	50.0 (6)

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
