# Peer review of "Clinical Characteristics of Mitochondrial DNA Copy Number in Osteonecrosis of the Femoral Head"

_medicina, 2020, doi:10.3390/medicina56050239_

Round 1
Reviewer 1 Report
The review would like to thank the authors for their efforts in this research and writing the manuscript. However, a few points are addressed to the authors:
The manuscript needs English language editing service. As an example, please check line 32, 34,....
INTRODUCTION
Line 36, Genetic study about mitochondrial diseases: Genetic studies.... and please add the references about those studies
Line 39, enhanced generation of adenosine triphosphate (ATP) by glycolysis. These studies suggest the clinical value of mtDNA-CN changes in various diseases [4–6].: I would continue this sentences as part of the former sentence and write it " by glycolosis suggesting the clinical value.....
Line 41, mtDNA-related issues: What kind of issues, is it mutation or what kind of deficiency?
Line 44-45, Many etiological factors, such as corticosteroids, dringking, radiation, and Gaucher disease, are associated with it development [8,9]: Drinking instead of dringking and I would prefer to write alcohol consumption than drinking as it's more scientific. English language editing is required.
MATERIALS AND METHODS
The M&M should be written in the passive form instead of writing the researchers everytime.
The table for the Primers sequence is missing.
RESULTS
Line 91-94, The mean age of the 157 patients with ONFH or femoral head fracture was 56.2 years (range,26–93 years). There were 70 (44.6%) male patients and 87 (55.4%) female patients. According to the clinical and pathological diagnosis, there were 34 (21.7%) ONFH patients and 123 (78.3%) femoral head fracture patients: How many male and female patient were present in ONFH and Fracture group. Would you please add it?
Line 117-118, Figure 2. A positive correlation between age and mitochondrial copy number in osteonecrosis of the
118 femoral head (A) and fracture (B): The authors had mentioned here that there is a positive correlation. However, in the discussion part, they had mentioned that the positive correlation was present only in the ONFH group. This means that the figure legend needs to be corrected. I would suggest that the legend would be changed to correlation only and not positive correlation.
The big question here is: What is the rational of mtDNA in the determination of various diseases if there was no significant difference between the ONFH and fracture group. Both have nearly the same mtDNA ct. Then mtDNA ct can be used only to differentiate between age groups in the sONFH group.
The mtDNA is not a precise method to differentiate between ONFH and other conditions. The only difference is the age which could be expected. Did you try to have a group of the same age and examine the mtDNA. Maybe you could find a difference?
Author Response
The review would like to thank the authors for their efforts in this research and writing the manuscript. However, a few points are addressed to the authors:
The manuscript needs English language editing service. As an example, please check line 32, 34,....
-> English language editing was performed.
INTRODUCTION
Line 36, Genetic study about mitochondrial diseases: Genetic studies.... and please add the references about those studies
Line 39, enhanced generation of adenosine triphosphate (ATP) by glycolysis. These studies suggest the clinical value of mtDNA-CN changes in various diseases [4–6].: I would continue this sentences as part of the former sentence and write it " by glycolosis suggesting the clinical value.....
-> It was revised.
Line 41, mtDNA-related issues: What kind of issues, is it mutation or what kind of deficiency?
-> All kind of genetic change of mtDNA was included. It was revised.
Line 44-45, Many etiological factors, such as corticosteroids, dringking, radiation, and Gaucher disease, are associated with it development [8,9]: Drinking instead of dringking and I would prefer to write alcohol consumption than drinking as it's more scientific. English language editing is required.
-> It was revised.
MATERIALS AND METHODS
The M&M should be written in the passive form instead of writing the researchers everytime.
The table for the Primers sequence is missing.
-> The passive form was revised. And Primer sequence was provided by reference.
RESULTS
Line 91-94, The mean age of the 157 patients with ONFH or femoral head fracture was 56.2 years (range,26–93 years). There were 70 (44.6%) male patients and 87 (55.4%) female patients. According to the clinical and pathological diagnosis, there were 34 (21.7%) ONFH patients and 123 (78.3%) femoral head fracture patients: How many male and female patient were present in ONFH and Fracture group. Would you please add it?
-> It was added.
Line 117-118, Figure 2. A positive correlation between age and mitochondrial copy number in osteonecrosis of the 118 femoral head (A) and fracture (B): The authors had mentioned here that there is a positive correlation. However, in the discussion part, they had mentioned that the positive correlation was present only in the ONFH group. This means that the figure legend needs to be corrected. I would suggest that the legend would be changed to correlation only and not positive correlation.
->The legend was revised.
The big question here is: What is the rational of mtDNA in the determination of various diseases if there was no significant difference between the ONFH and fracture group. Both have nearly the same mtDNA ct. Then mtDNA ct can be used only to differentiate between age groups in the sONFH group.
The mtDNA is not a precise method to differentiate between ONFH and other conditions. The only difference is the age which could be expected. Did you try to have a group of the same age and examine the mtDNA. Maybe you could find a difference?
->The weak significance of mtDNA was the limitation of this study. Therefore, it was described in Discussion part. IRB provided the patients group considering patients age. Therefore, it did not affect the result. It was described in Result.
Reviewer 2 Report
Dear authors, dear Editor,
Thank you for the opportunity to review this work.
The topic seems to me very interesting, original and that opens new research perspectives for a rare but important adverse event.
In my opinion this work can be published.
Author Response
Thank you for your kind review and reply.
-------------------------------------------
Dear authors, dear Editor,
Thank you for the opportunity to review this work.
The topic seems to me very interesting, original and that opens new research perspectives for a rare but important adverse event.
In my opinion this work can be published.
-------------------------------------------